# AutoST: Towards the Universal Modeling of Spatio-temporal Sequences

**Jianxin Li**[*]
BDBC[†]
Beihang University
Beijing, China 100191
lijx@buaa.edu.cn

**Shuai Zhang**
BDBC
Beihang University
Beijing, China 100191
zhangs@act.buaa.edu.cn

**Hui Xiong**
HKUST(GZ)[‡]
HKUST FYTRI[§]
Guangzhou, China 511455
xionghui@ust.hk

**Haoyi Zhou**
BDBC
Beihang University
Beijing, China 100191
haoyi@buaa.edu.cn

## Abstract

The analysis of spatio-temporal sequences plays an important role in many real-world applications, demanding a high model capacity to capture the interdependence among spatial and temporal dimensions. Previous studies provided separated network design in three categories: spatial first, temporal first, and spatio-temporal synchronous. However, the manually-designed heterogeneous models can hardly meet the spatio-temporal dependency capturing priority for various tasks. To address this, we proposed a universal modeling framework with three distinctive characteristics: (i) Attention-based network backbone, including S2T Layer (spatial first), T2S Layer (temporal first), and STS Layer (spatio-temporal synchronous). (ii) The universal modeling framework, named UniST, with a unified architecture that enables flexible modeling priorities with the proposed three different modules. (iii) An automatic search strategy, named AutoST, automatically searches the optimal spatio-temporal modeling priority by network architecture search. Extensive experiments on five real-world datasets demonstrate that UniST with any single type of our three proposed modules can achieve state-of-the-art performance. Furthermore, AutoST can achieve overwhelming performance with UniST.

## 1 Introduction

Modeling and predicting the future of spatio-temporal (ST) sequences based on past observations has been extensively studied and has been successfully applied in many fields, such as road traffic [13], medical diagnosis [29], and meteorological research [24]. Traditional statistical methods typically require input sequence satisfying certain assumptions, which limits its ability in capturing the complex spatial-temporal dependency. Then, recurrent neural network (RNN) methods [15] leverage the universal approximation property to build separated network branches to model dependency and make predictions with fusion gate blocks from the stacking branches. The intrinsic gradient flow in the back-propagation training process [7] may bring the ST dependency into incorrespondence with the

---

[*]Jainxin Li is the corresponding author.
[†]BDBC: Beijing Advanced Innovation Center for Big Data and Brain Computing.
[‡]HKUST(GZ): Hong Kong University of Science and Technology (Guangzhou).
[§]HKUST FYTRI: Guangzhou HKUST Fok Ying Tung Research Institute.

36th Conference on Neural Information Processing Systems (NeurIPS 2022).

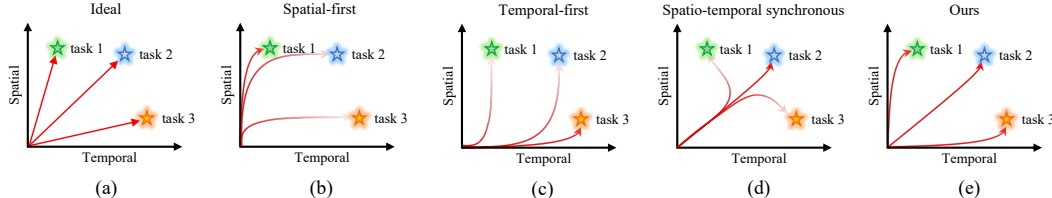

(a) Ideal    (b) Spatial-first    (c) Temporal-first    (d) Spatio-temporal synchronous    (e) Ours

Figure 2: Modeling orders of ST data. The stars are tasks with different spatial-temporal dependencies. The red lines are spatio-temporal modeling procedure with anisotropic tendency. The red lines' color going darkness/lightness refers to the modeling ability of the model along with the current modeling tendency.

network branches' configuration, especially for the deeper network [11]. Recently, the Transformer-based models show larger modeling capacity in both spatial and temporal modeling [21, 32, 31], which motivates us to find a universal way to capture the ST dependency in a universal framework simultaneously.

As shown in Fig. 1, the ST dependency individually exists in sequences: spatial correspondences, temporal correspondences, and spatio-temporal correspondences. Take the road traffic forecasting as an example, previous research fall into three typical paradigms. (a) **Spatial-first modeling** [1, 27, 13]: Predicting the traffic for next road junction, which has strong connec-

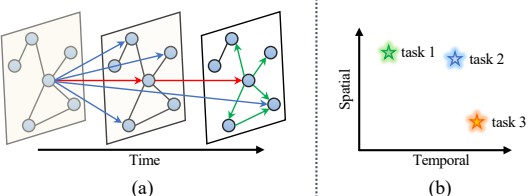

Figure 1: (a) Dependencies in ST sequence data: Green arrows are spatial dependencies; Red arrows are temporal dependencies; Blue arrows are dependencies across both spatial and temporal. (b) Task 1 has high spatial dependency but low temporal dependency; Task 3 has high temporal dependency but low spatial dependency; Task 2 has equal spatial and temporal dependencies.

tions to previous intersections, stops, and surrounding traffic. (b) **Temporal-first modeling** [23, 5]: Predicting the traffic of a high school on Friday afternoon, which shows strong periodic relationships with the school's times schedule. (c) **Spatio-temporal synchronous modeling** [19, 12, 28, 6]: Analyzing the city-wide traffic, which is tightly entangled in both spatial and temporal. During the sequences modeling, the main problem is to align the network design with the natural spatio-temporal distribution.

However, the distribution of ST dependencies varies and depends on the forecasting task and corresponding datasets. They are mixed in a compound way when modeling ST sequences, and the three tasks in Fig. 1(b) are the representative ones. What makes it worse is that, the prevalent modeling methods show anisotropic tendency to capture the ST dependency. If we use the spatial-first models on the three tasks in Fig. 2(b), the task 1's states are highly influenced by the surrounding information, and the periodic pattern is the underlying factors, which makes the spatial-first model fits it properly. We can compare the model ability (red lines) with the ideal one in Fig. 2(a), this kind of model will be insufficient for task 2 and task 3. Similarly, suppose we use the temporal-first models on the three tasks in Fig. 2(c). In that case, the model ability only matches task 3, where the periodic pattern decides the states other than the spatial information. The previous analysis also applies to the spatio-temporal synchronous situation, where the states are mainly influenced by the complex associations across the spatial and temporal, like semantic relationships. In this paper, we aim to propose a universal model that alleviates the the modeling gap on different tasks.

The contributions are: 1) The first to raise and address the modeling order problem in spatio-temperal forecasting tasks by proposing a universal modeling framework UniST and an automatic structure search strategy AutoST. 2) Proposing 3 replacable and unified attention-based modeling units named S2T, T2S and STS, which model spatio-time sequence with three different priorities: spatial first, temporal first and spatio-temperal synchronous. 3) Extensive experiments on 5 datasets and 3 sequence forcasting tasks demonstrate that only using our three modeling units (S2T, T2S, and STS) outperforms the baseline methods, and our framework together with AutoST achieves the new state-of-the-art performance.

## 2    Related Work

Existing spatio-temporal forecasting methods can be roughly grouped into three categories: spatial-first, temporal-first, and spatio-temporal synchronous methods. **Spatial-first**: STG2Seq [1] uses stacking GCN layers to capture the entire inputs sequence, where each GCN layer operates on a limited historical time window, and the final results are concatenated together to make forecasting. In the view of this paper, it belongs to spatial-first modeling. STGCN [27] propose the blocks that contains two temporal gated convolution layers with one spatial graph convolution layer in the middle, which starts from a convolution-based temporal layer. DCRNN [13] is proposed to forecast traffic flow using diffusion convolution and recurrent units to capture spatial and temporal information successively. **Temporal-first**: Graph WaveNet [23] built the basic modeling layers with two gated temporal convolution modules at the beginning and followed by a graph convolution module, which models from temporal to spatial. GSTNet [5] builds several layers of spatial-temporal blocks to produce the forecasting, which is consists of a multi-resolution temporal module followed by a global correlated spatial module. **Spatio-temporal synchronous**: STSGCN [19] construct a spatio-temporal synchronous extraction module composed of graph convolutional networks. STFGNN [12] modeling spatio-temporal correlations simultaneously by fusing a dilated convolutional neural network with a gating mechanism and a spatio-temporal fusion graph module. ST-ResNet [28] using convolution on a sequence of image-like 2D matrices to model spatio-temporal at the same time. ASTGCN [6] proposed a spatial-temporal convolution that simultaneously captures the spatial patterns and temporal features.

## 3    Preliminary

### 3.1    Spatio-temporal Sequence Forecasting

Spatio-temporal sequence forecasting (STSF) is to predict the future sequence of spatio-temporal inputs based on the historical observations. Specifically, given a graph $G = (V, E, A)$, where $V$ and $E$ are the node set and edge set, and $N$ is the number of nodes, $A \in \mathbb{R}^{N \times N}$ is the adjacency matrix of $G$. If $v_i, v_j \in V$ and $(v_i, v_j) \in E$, $A_{ij} = 1$, otherwise $A_{ij} = 0$. $\mathbf{X} = \{\mathbf{X}_1, \mathbf{X}_2, \ldots, \mathbf{X}_T\}$ is a ST sequence of $T$ time steps, where $\mathbf{X} \in \mathbb{R}^{T \times N \times C}$. The snapshot at time step $t$ is denoted as $\mathbf{x}_t \in \mathbb{R}^{N \times C}$ , where $C$ is the feature dimension of a node. Then the ST sequence forecasting problem can be defined as: given $S$ time steps historical observations of input graph $G$, the goal is to predict the future sequence of the features on each node with a learning function $f$: $\left[\mathbf{X}_{(t-S):t}, G\right] \xrightarrow{f} \mathbf{X}_{(t+1):(t+P)}$, where $\mathbf{X}_{(t-S):t}$ and $\mathbf{X}_{(t+1):(t+P)}$ are the ST sequence with length $S$ and $P$ respectively.

### 3.2    Network Architecture Search

Network (neural) architecture search (NAS) are automated methods for generating and optimizing neural networks. A representative gradient-based approach is DARTS [16], which is the foundation of our proposed training framework. DARTS aims to search optimal directed edge connections on a directed acyclic graph with predefined computing cells as nodes. The result connections of node $j$ is denoted as $x^{(j)} = \sum_{i<j} o^{(i,j)}\left(x^{(i)}\right)$, where $o^{(i,j)}$ is an operator, e.g. layers in a model, represented by a directed edge from node i to node j. DARTS proposes a method to relax the discrete searching space to be continuous, and uses bi-level optimization to learn a differentiable objective on the joint optimization problem of both network architecture and model weights. The objective function is: $\min_\alpha L_{val}\left(w^*(\alpha), \alpha\right)$ s.t. $w^*(\alpha) = \mathrm{argmin}_w L_{train}(w, \alpha)$, where $\alpha$ is the architecture, and $w$ is the model weights. In Section 4.4, we improve the design of the directed acyclic graph of the search architecture, and the two-stage optimization of the architecture parameters.

## 4    Methods

In this section, we firstly introduce two basic modeling units: the time series linear self-attention, and the high order mix graph convolution. Then we proposed three layers as different network backbones, and we build a universal modeling framework based on the tree "atomic" layers. Next, we propose an automatic searching strategy for spatio-temporal information fusion, which aimed for the optimal order of spatio-temporal modeling on various downstream tasks.

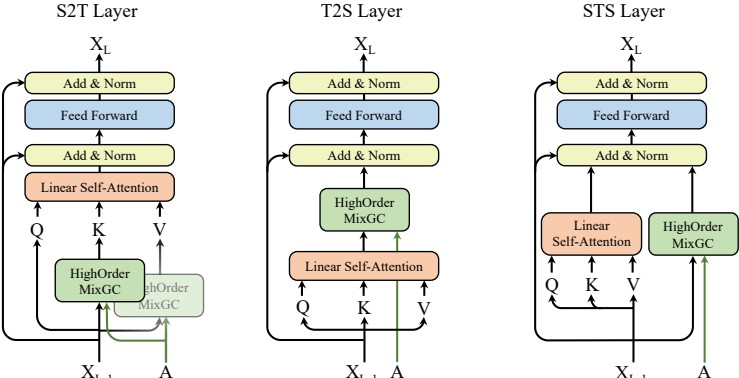

Figure 3: Three ST modeling modules of different modeling orders. (a) S2T Layer: firstly model spatial information, then combine with temporal information within self-attention mechanism; (b) T2S Layer: firstly model temporal information, then use the high-order graph convolution to capture the complex dependency; (c) STS Layer: model the spatial and temporal information simultaneously, and concatenate the two feature map as the final representation.

## 4.1 Spatial / Temporal Modeling Unit

### 4.1.1 Time Series Linear Self-Attention

Self-attention mechanism [21] has been widely used in nature language processing, computer vision, and time series forecasting, which is defined as: $\text{Attention}(\mathbf{Q}, \mathbf{K}, \mathbf{V}) = \mathbf{V}' = \text{Softmax}(\mathbf{Q}\mathbf{K}^\top/\sqrt{d})\mathbf{V}$, where $\mathbf{Q} = \mathbf{X}\mathbf{W}_Q, \mathbf{K} = \mathbf{X}\mathbf{W}_K, \mathbf{V} = \mathbf{X}\mathbf{W}_V$, and the projection matrix $\mathbf{W}_Q \in \mathbb{R}^{C \times D}, \mathbf{W}_K \in \mathbb{R}^{C \times D}, \mathbf{W}_V \in \mathbb{R}^{C \times D}$.

However, the original self-attention suffers from high computational and memory cost. Because the dot product computation of $\mathbf{Q}$ and $\mathbf{K}$ leads to $O(N^2)$ time and space complexity. [9] proposed linear self-attention, which represents the similarity function of $\mathbf{Q}$ and $\mathbf{K}$ in the self-attention by a kernel function: $\mathbf{V}'_i = \phi(\mathbf{Q}_i)^T \sum_{j=1}^{N} \phi(\mathbf{K}_j) \mathbf{V}_j^T / \phi(\mathbf{Q}_i)^T \sum_{j=1}^{N} \phi(\mathbf{K}_j)$. Such that for each query $\mathbf{Q}_i$, the two terms $\sum_{j=1}^{N} \phi(\mathbf{K}_j)\mathbf{V}_j$ and $\sum_{j=1}^{N} \phi(\mathbf{K}_j)$ are the same and reused for efficient computing. Following [31], we use the technique of linear self-attention in representing time series features.

### 4.1.2 High-order Mix Graph Convolution

To acquire better spatial information representation, we propose a high-order mix graph convolutional operation for spatial information mixing and feature extraction of the original inputs, it is defined as:

$$\text{HighOrder}(\mathbf{X}, \mathbf{A}, order)$$
$$\stackrel{def}{=} \mathbf{H}^{order} = \begin{cases} \mathbf{X} & if \ order = 0 \\ \text{MixGC}(\mathbf{X}, \mathbf{A}) & if \ order = 1 \\ \text{MixGC}(\mathbf{H}^{(order-1)}\mathbf{A}) & if \ order > 1, \end{cases} \tag{1}$$

where $order$ denotes the total order of the graph convolution operations, i.e., to consider $order$-hop neighbor relationship of each node. In this paper, we define the $1^{st}$-order mix convolutional operation by combining the $1^{st}$-order ChebNet [10] and the Adaptive Diffusion Convolution [23]: $\text{MixGC}(\mathbf{X}, \mathbf{A}) = \text{ChebNet}(\mathbf{X}, \mathbf{A}) + \text{AdapDC}(\mathbf{X}, \mathbf{A}) = \hat{\mathbf{A}}\mathbf{X}\mathbf{W}_g + \mathbf{P}_f\mathbf{X}\mathbf{W}_f + \mathbf{P}_b\mathbf{X}\mathbf{W}_b + \hat{\mathbf{A}}_{adp}\mathbf{X}\mathbf{W}_{adp}$, where $\hat{\mathbf{A}} = \mathbf{D}^{-1/2}\tilde{\mathbf{A}}\mathbf{D}^{-1/2}$ is a normalized adjacency matrix with self-loop. ChebNet focuses on 1st-order neighbor information, while AdapDC focuses on multi-hop information. $\tilde{\mathbf{A}}$ is defined as $\tilde{\mathbf{A}} = \mathbf{A} + \mathbf{I}$, where $\mathbf{D}_{ii} = \sum_j \tilde{\mathbf{A}}_{ij}$, $\mathbf{I}$ is an identity matrix. $\mathbf{P}_f = \frac{\mathbf{A}}{\text{rowsum}(\mathbf{A})}$, $\mathbf{P}_b = \frac{\mathbf{A}^\top}{\text{rowsum}(\mathbf{A}^\top)}$ refers to a forward and backward state transition matrix, respectively. $\hat{\mathbf{A}}_{adp}$ is an adaptive matrix for complementary spatial state information, which is calculated by two learnable node embedding matrices $\mathbf{E}_1, \mathbf{E}_2 \in \mathbb{R}^{N \times C}$ [20] as $\hat{\mathbf{A}}_{adp} = \text{Softmax}(\text{ReLU}(\mathbf{E}_1\mathbf{E}_2^\top))$.

## 4.2 Unified Spatio-temporal Modeling Backbone

In order to solve the problem of spatio-temporal dependency distribution differences in the modeling procedure, we first propose three novel modules: *S2T Layer*, *T2S Layer*, *STS Layer*, that are suitable for three typical spatio-temporal dependencies: spatial-first, temporal-first, spatio-temporal synchronous, respectively. We design all these three modeling module to have the same dimension of inputs and outputs. This provides a solid foundation for our later flexible and universal modeling.

### 4.2.1 Spatial-first Modeling Layer

The spatial-first sequence modeling method, *S2T Layer*, models from spatial to temporal. The spatial information between the nodes on the graph is first characterized on a single slice. After that, node information at different times is exchanged along the time dimension, whose spatial information has been shared with its neighbors.

As shown in Fig.3(a), *S2T Layer* first uses two high-order mix graph convolution defined in Eq.(1) to process the input spatio-temporal sequences $\mathbf{X}_{L-1}$ to obtain two sequence representations with mixed spatial information. Then the key $\mathbf{K}$ and value $\mathbf{V}$ of the input of the subsequent self-attention are obtained by a transformation using the parameter matrix $\mathbf{W}_K, \mathbf{W}_V$, respectively, while the query $\mathbf{Q}$ is obtained by transforming the original input ST sequence using the parameter matrix as follows: $\mathbf{Q} = \mathbf{X}_{L-1}\mathbf{W}_Q, \mathbf{K} = \text{HighOrder}_1(\mathbf{X}_{L-1}, \mathbf{A}, order)\mathbf{W}_K, \mathbf{V} = \text{HighOrder}_2(\mathbf{X}_{L-1}, \mathbf{A}, order)\mathbf{W}_V$.

Then the original ST sequence and the new sequence with mixed spatial information are processed using a multi-head linear self-attention, from which it learns temporal dependencies and exchanges information at different time slices to obtain further representations of the ST sequence:

$$\mathbf{Z} = \text{Attention}(\mathbf{Q}, \mathbf{K}, \mathbf{V}) \quad . \tag{2}$$

The output is concatenated with the initial input once for residuals and processed a layer normalization, followed by a two-layer fully connected network for further ST representation learning. This network is applied separately and identically to each point-in-time position in the ST sequence, thus maintaining the continuous transfer of position-encoded information. Finally, the resulting ST sequence representation is again connected to the initial input with one residual and layer normalization to obtain the output $\mathbf{X}_L = \text{Norm}(\max(0, \text{Norm}(\mathbf{Z} + \mathbf{X}_{L-1})\mathbf{W}_1 + b_1)\mathbf{W}_2 + b_2)$.

### 4.2.2 Temporal-first Modeling Layer

This module is designed to model the ST sequence from temporal to spatial, named *T2S Layer*. Different from spatial-first modeling, at the beginning, the original inputs are projected into $\mathbf{Q}, \mathbf{K}, \mathbf{V}$ by three weight matrices as $\mathbf{Q} = \mathbf{X}_{L-1}\mathbf{W}_Q, \mathbf{K} = \mathbf{X}_{L-1}\mathbf{W}_K$, and $\mathbf{V} = \mathbf{X}_{L-1}\mathbf{W}_V$. The projection results are used to calculate temporal representations at first, using the time series linear self-attention in Eq.(2). Then the temporal representation on each node are send to the high-order mix graph convolution, together with the adjacency matrix, to fusion the temporal information from every neighbors. $\mathbf{Z}' = \text{HighOrder}(\mathbf{Z}, \mathbf{A}, order)$. Finally, the output representations are executed with the feed forward and layer normalization operations the same way as the *S2T Layer*.

### 4.2.3 Spatial-temporal Synchronous Layer

This module named *STS Layer*, which aims to model the spatial and temporal information simultaneously. Different from the former two modules, the inputs are directly used to calculate spatial and temporal representations at the same time. For temporal modeling part, it still project the original inputs into $\mathbf{Q}, \mathbf{K}, \mathbf{V}$, and execute a linear self-attention operation for a temporal representation. For the spatial part, it accepts the original inputs and the spatial information and uses high-order mix graph convolution operation to construct spatial representation. $Temporal \leftarrow \mathbf{Z}_1 = \text{Attention}(\mathbf{Q}, \mathbf{K}, \mathbf{V}), Spatial \leftarrow \mathbf{Z}_2 = \text{HighOrder}(\mathbf{X}_{L-1}, \mathbf{A}, order)$. The outputs are concatenated together as: $\mathbf{Z}' = \text{concat}[\mathbf{Z}_1, \mathbf{Z}_2]$. Then it is executed with the following operations and output as $\mathbf{X}_L$ similar with the former two modules.

## 4.3 Universal Modeling Framework

Targeting to the ST sequence forecasting task, we propose a unified ST sequence modeling framework (UniST) with the proposed unified modeling backbones in Fig.4, which follows the encoder-decoder architecture. It uses a unified architecture with interchangeable and replaceable mode units.

### 4.3.1 Spatio-temporal Embedding Layer

Since the Transformer model solely relies on the self-attention for global alignments, the positional embedding [21] and extra embeddings [32] are needed to capture spatio-temporal dependency. Then we introduce four types of embeddings $\mathbf{E}_P$, $\mathbf{E}_V$, $\mathbf{E}_S$, $\mathbf{E}_T$ in Appendix A.

**Fusion embedding**. The four embeddings are summed together as the final embedding added to the inputs: $\mathbf{E}_F = \mathbf{E}_P + \mathbf{E}_V + \mathbf{E}_S + \mathbf{E}_T$, note that the shape of token embedding $\mathbf{E}_V \in \mathbb{R}^{T \times N \times d}$, while other embeddings' shape are $\mathbf{E}_P \in \mathbb{R}^{1 \times 1 \times d}$, $\mathbf{E}_S \in \mathbb{R}^{1 \times N \times d}$, $\mathbf{E}_T \in \mathbb{R}^{T \times 1 \times d}$. When calculating the summation, they will be replicated and expanded with broadcast on the respective missing dimensions.

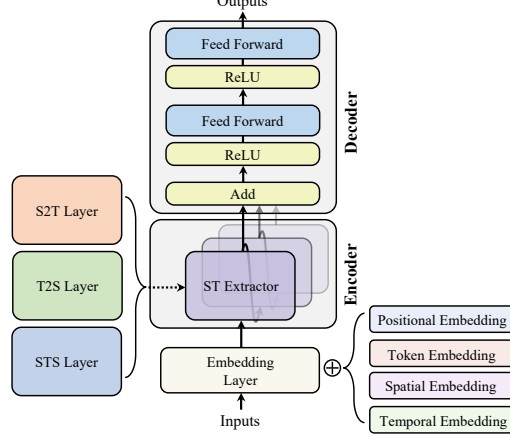

Figure 4: Overview of the UniST framework, starting with an embedding layer and followed by an encoder and a decoder. The embedding layer generates the sum of four embeddings. The encoder consists of multiple spatio-temporal extractors, including S2T, T2S or STS Layer.

### 4.3.2 Encoder

The encoder of UniST consists of multiple Spatio-Temporal Extractors (STE($\cdot$)), which can be arbitrarily chosen from {*T2S Layer*, *S2T Layer*, *STS Layer*}. All extractors are connected end to end, i.e., the output of the previous one is the input of the next one. To acquire a more diversity representation, the outputs of each extractor are added to form the final output of the encoder. Let the outputs of the embedding layer be $\mathbf{X}_0$, the encoder is computed as: $\mathbf{X}_{en} = \sum_{i=1}^{L} \text{STE}^i(\mathbf{X}_0)$, where $L$ refers to the number of spatio-temporal extractors.

### 4.3.3 Decoder

The decoder accepts the output of encoder, i.e., $L$ outputs from $L$ spatio-temporal extractors. They are firstly added as a unified spatio-temporal representation. Then the results are through two times of ReLU activation and Linear projection, and produce the final sequence forecasting result. Denote $\mathbf{X}_\ell$ as the output of extractor $\ell$, we have the calculation of decoder as: $\bar{\mathbf{Y}} = \mathbf{X}_{de} = \text{Linear}(\text{ReLU}(\text{Linear}(\text{ReLU}(\sum_\ell \mathbf{X}_\ell))))$.

### 4.4 Automated Search for UniST

With the proposed unified ST sequence modeling framework UniST, it still suffers from the potential wrong network configuration problem, where we build an arbitrary modeling order with the replaceable model units {*T2S Layer*, *S2T Layer*, *STS Layer*}. Considering the various downstream tasks, how can we build a universal model with an optimal configuration? Here we propose the Automated Spatio-Temporal modeling approach (AutoST), which learns the optimal combinatorial order that suits the spatio-temporal dependency of the current task. We designed two schemes for layer combination. In this section, we first define the basic searching unit of AutoST, then we introduce two designs of AutoST with different searching schemes.

### 4.4.1 AutoST Cell

The basic searching unit in AutoST is the network cell. Here we define its structure and computing process.

**Definition 1. *AutoST Cell.*** *Let $\mathcal{G} = (\mathcal{V}, \mathcal{E})$ be a direct acyclic graph (DAG), $\mathcal{V}$ denotes the node set, each node refers to a representation comes from the outputs of a computation layer. The representation on node $i$ is defined as $\mathbf{H}^i \in \mathbb{R}^{T \times |\mathcal{V}| \times d}$, where $T$ is the length of spatio-temporal seqence, $d$ is the*

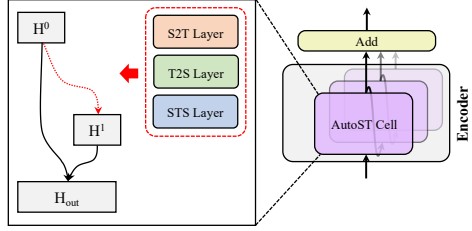 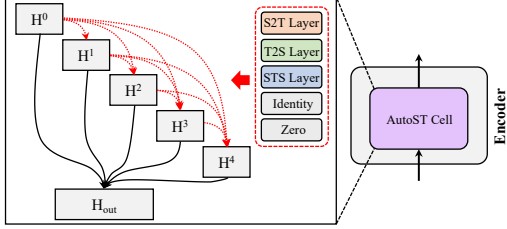

(a) AutoST$_1$: the sequential stacking scheme.     (b) AutoST$_2$: the hybrid assembling scheme.

Figure 5: Two AutoST searching schemes. (a) AutoST$_1$ is the sequential stacking scheme. Only search for one of the three modeling layers to build the AutoST Cell. The cells are stacking sequentially to form the encoder, and each cell is skip-connected. (b) AutoST$_2$ is the hybrid assembling scheme, with only one AutoST Cell in the encoder. The cell consists of multiple nodes for intermediate representations. The connections between nodes are candidate layers.

*feature dimension. The input of each AutoST Cell is denoted as $\mathbf{H}^0$, and the output of each cell is the summation of all nodes, i.e., all interval representations: $\mathbf{H}_{out} = \sum_i^{|\mathcal{V}|} \mathbf{H}^i$ . On graph $\mathcal{G}$, the directed edge $(i,j)$ from node $i$ to node $j$ stands for a mixture of all candidate modeling modules $O = \{T2S\ Layer,\ S2T\ Layer,\ STS\ Layer\}$, and it is represented as $o^{(i,j)}$. So that the representation between node $j$ and other nodes can be written as: $\mathbf{H}_j = \sum_{i<j} o^{(i,j)} \mathbf{H}^i$. On each directed edge, there exist a set of weight parameters $\alpha^{(i,j)} = \{\alpha_o^{(i,j)} | o \in O\}$, which indicates the probability of the corresponding modeling module should be retained. Every weight parameters of the candidate modeling module is calculated as: $\mathbf{H}^j = \sum_{i<j} \sum_{o \in O} \frac{\exp\left(\alpha_o^{(i,j)}\right)}{\sum_{o' \in O} \exp\left(\alpha_{o'}^{(i,j)}\right)} o\left(\mathbf{H}^i\right).$*

### 4.4.2 Sequential Stacking Search

Based on the proposed UniTS, we propose two searching schemes to find better combination of the modeling layers. The proposed searching schemes are lossless replacements for the encoder of UniTS. We simply replace the encoder' spatio-temporal extractor with the AutoST Cell.

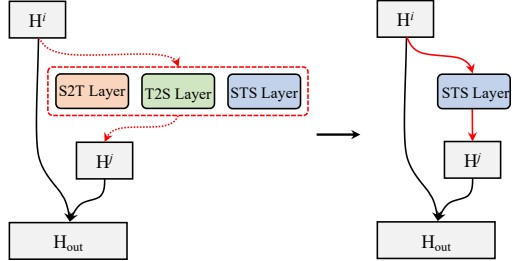

The first one is multi-layer sequential stacking searching scheme, with which the whole model is named AutoST$_1$. As illustrated in Fig.5(a), it has simple structure within each AutoST Cell, while has more complicated stacking structure between cells. Each cell holds a DAG with three nodes,

Figure 6: Searching process of the AutoST Cell. The searching goal is to choose the optimal one from the candidate modules between the given computation node $\mathbf{H}^i$ and $\mathbf{H}^j$.

one input node $\mathbf{H}^0$, one output node $\mathbf{H}_{out}$ and an intermediate node $\mathbf{H}^1$. And two directed edges are pre-defined between the former two nodes' output and the output node: $(\mathbf{H}^0, \mathbf{H}_{out})$ and $(\mathbf{H}^1, \mathbf{H}_{out})$.

The searching space of this scheme is shown as the directed red dotted line. The red dashed box is the search candidate set, including {*T2S Layer*, *S2T Layer*, *STS Layer*}. We conduct two gradient based network architecture search methods in the experiments, i.e., DARTS [16], PAS [22]. After searching, the cell essentially becomes one of the three model units. This scheme allows multiple stacking of cells, therefore, the new encoder of the whole model will become the sequential stacking between different modeling layers.

### 4.4.3 Hybrid Assembling Search

The another searching scheme is called hybrid assembling searching. The structure is similar with the sequential stacking searching. However, in this scheme, the encoder only consists of a single AutoST Cell. The cells are not stacked layer-by-layer, it will conduct searching on a more complicated DAG on a single AutoST Cell.

In this searching scheme, the DAG of AutoST Cell is shown as Fig.5(b). There are multiple nodes in the cell, generally, it will be set as 4-7 nodes in the experiments. The pre-defined connections are

each node's output to the output node. The multiple directed red dotted lines show the searching space of this scheme. The candidate set is expanded with two operations: *Identity* and *Zero*. *Identity* means no modeling module in this edge, i.e., build directly connection with no operation. *Zero* means to set all the data passing through it to zero, i.e., no connection is made. This scheme makes the entire DAG form a complex and deep network structure. Through the complex structure design inside the cell, multiple spatio-temporal modeling modules are combined to fit the spatio-temporal dependency distribution in a target task.

# 5    Experiments

This section empirically evaluates the effectiveness of UniST and AutoST models with short-term, medium-term, and long-term ST sequence forecasting tasks on five real-world datasets. Platform: Intel(R) Xeon(R) CPU  2.40GHz $\times$ 2 + NVIDIA Tesla V100 GPU (32 GB) $\times$ 4. The code is available at https://github.com/shuaibuaa/autost2022.

## 5.1    Datasets

In order to study the effect of various ST sequence forecasting methods under complex spatio-temporal distribution, five real-world datasets with different tasks and data states are selected. The statistic information of the five datasets are listed in Table 3.

**METR-LA** [8]: The traffic speed dataset contains 4 months of data from March 1, 2012 to June 30, 2012, recorded by sensors at 207 different locations on highways in Los Angeles County, USA. The data granularity is 5 minutes per point, and the spatial information provided by the dataset includes the coordinates of each sensor and the distance between the sensors.

**PEMS-BAY** [14]: The traffic speed dataset comes from the California Transportation Agencies (CalTrans) Performance Evaluation System (PeMS). The dataset contains data recorded by 325 sensors in the Bay area for a total of 6 months from January 1, 2017 to May 31, 2017, and the data granularity is 5 minutes per point. The spatial information provided by the dataset includes the coordinates of each sensor and the distance between the sensors.

**PEMS-03/04/08** [3]: The three traffic datasets are also from the PeMS system of the California Transportation Agency, and each dataset is data recorded by sensors in a certain area of California. The PEMS-03 dataset contains data recorded by 358 sensors for 3 months from September 1, 2018 to November 30, 2018. PEMS-04 dataset contains 2 months of data recorded by 307 sensors from January 1, 2018 to February 28, 2018. PEMS-08 dataset contains 2 months of data recorded by 170 sensors from July 1, 2016 to August 31, 2016. The data granularity of these three datasets is 5 minutes per point, and the spatial information provided by the datasets only includes the connectivity between sensors.

## 5.2    Main Results

Table 1 summarizes the ST sequence forecasting results. $UniST_S$, $UniST_T$, $UniST_{ST}$ stands for our UniST framework with three same stacking layers of *S2T Layer*, *T2S Layer*, *STS Layer*, respectively.

The results in bold font in Table 1 show that our proposed UniST outperforms all baseline methods and achieves State-of-the-Art on all 9 tasks of 5 datasets, with all three proposed layers. This demonstrates our proposed unified spatio-temporal modeling layers and the unified forecasting framework are more expressive than traditional methods. Specifically, compared with GMAN, which is also based on self-attention mechanism, our methods achieve at most 18.41%, 15.31%, 13.82% MAE decreases on the short-term, medium-term, and long-term forecasting task, respectively. Compared with the most advanced method STFGNN, our methods gain 12.06%, 10.61%, 9.43% MAE decreases on the short, medium, long-term forecasting tasks.

From the last two columns of Table 1, we can see that both of $AutoST_1$ and $AutoST_2$ beat all other methods on every metric of every task, including our proposed UniST. Recall that the difference between AutoST and UniST is the encoder part, UniST uses all same layers from {*S2T Layer*, *T2S Layer*, *STS Layer*}, while AutoST aims to search for a better combination and connection using the three types of layers. This demonstrates that combining and integrating modules with different spatio-temporal modeling abilities can better deal with uncertainty spatio-temporal dependency and

Table 1: Spatio-temporal sequence forecasting performance. The bold and shaded numbers are the best results of all methods. The bold numbers are the best results of manually designed models.

| Dataset | | Metric | VAR | SVR | ARI. | WAV. | DCR. | STG. | G.WV. | STF. | GMA. | UniST$_S$ | UniST$_T$ | UniST$_{ST}$ | AutoST$_1$ | AutoST$_2$ |
|---|---|---|---|---|---|---|---|---|---|---|---|---|---|---|---|---|
| METR-LA | 15 min | RMSE | 7.89 | 8.45 | 8.21 | 5.89 | 5.38 | 5.74 | 5.15 | 4.73 | 5.48 | **4.39** | 4.43 | 4.47 | 4.38 | **4.33** |
| | | MAE | 4.42 | 3.99 | 3.99 | 2.99 | 2.77 | 2.88 | 2.69 | 2.57 | 2.77 | **2.25** | 2.29 | 2.33 | 2.23 | **2.19** |
| | | MAPE | 10.20 | 9.30 | 9.60 | 8.04 | 7.30 | 7.62 | 6.90 | 6.51 | 7.25 | **5.63** | 5.75 | 5.89 | 5.58 | **5.51** |
| | 30 min | RMSE | 9.13 | 10.87 | 10.45 | 7.28 | 6.45 | 7.24 | 6.22 | 5.46 | 6.34 | 5.20 | **5.12** | 5.22 | 5.09 | **5.02** |
| | | MAE | 5.41 | 5.05 | 5.15 | 3.59 | 3.15 | 3.47 | 3.07 | 2.83 | 3.07 | 2.57 | **2.53** | 2.61 | 2.50 | **2.44** |
| | | MAPE | 12.70 | 12.10 | 12.70 | 10.25 | 8.80 | 9.57 | 8.37 | 7.46 | 8.35 | 6.88 | **6.76** | 6.97 | 6.64 | **6.54** |
| | 60 min | RMSE | 10.11 | 13.76 | 13.23 | 8.93 | 7.60 | 9.40 | 7.37 | 6.40 | 7.21 | 6.09 | **6.07** | 6.23 | 6.03 | **6.00** |
| | | MAE | 6.52 | 6.72 | 6.90 | 4.45 | 3.60 | 4.59 | 3.53 | 3.18 | 3.40 | 2.91 | **2.89** | 3.02 | 2.85 | **2.78** |
| | | MAPE | 15.80 | 16.70 | 17.40 | 13.62 | 10.50 | 12.70 | 10.01 | 8.81 | 9.72 | 8.17 | **8.12** | 8.47 | 7.88 | **7.79** |
| PEMS-BAY | 15 min | RMSE | 3.16 | 3.59 | 3.30 | 3.01 | 2.95 | 2.96 | 2.74 | 2.33 | 2.82 | **2.26** | 2.27 | 2.32 | 2.21 | **2.16** |
| | | MAE | 1.74 | 1.85 | 1.62 | 1.39 | 1.38 | 1.36 | 1.30 | 1.16 | 1.34 | **1.11** | 1.12 | 1.14 | 1.09 | **1.07** |
| | | MAPE | 3.60 | 3.80 | 3.50 | 2.91 | 2.90 | 2.90 | 2.73 | 2.41 | 2.81 | **2.21** | 2.25 | 2.28 | 2.16 | **2.12** |
| | 30 min | RMSE | 4.25 | 5.18 | 4.76 | 4.21 | 3.97 | 4.27 | 3.70 | 3.02 | 3.72 | 2.94 | 2.96 | **2.92** | 2.89 | **2.85** |
| | | MAE | 2.32 | 2.48 | 2.33 | 1.83 | 1.74 | 1.81 | 1.63 | 1.39 | 1.62 | 1.33 | 1.36 | **1.32** | 1.29 | **1.27** |
| | | MAPE | 5.00 | 5.50 | 5.40 | 4.16 | 3.90 | 4.17 | 3.67 | 3.02 | 3.63 | 2.83 | 2.81 | **2.79** | 2.73 | **2.70** |
| | 60 min | RMSE | 5.44 | 7.08 | 6.50 | 5.43 | 4.74 | 5.69 | 4.52 | 3.74 | 4.32 | 3.66 | 3.67 | **3.64** | 3.60 | **3.58** |
| | | MAE | 2.93 | 3.28 | 3.38 | 2.35 | 2.07 | 2.49 | 1.95 | 1.66 | 1.86 | 1.62 | 1.63 | **1.61** | 1.57 | **1.52** |
| | | MAPE | 6.50 | 8.00 | 8.30 | 5.87 | 4.90 | 5.79 | 4.63 | 3.77 | 4.31 | 3.59 | 3.60 | **3.58** | 3.52 | **3.48** |
| PEMS3 | 60 min | RMSE | 38.26 | 35.29 | 34.98 | 33.65 | 30.31 | 30.12 | 32.94 | 28.34 | 33.21 | 26.75 | **26.61** | 27.09 | 26.26 | **25.71** |
| | | MAE | 23.65 | 21.97 | 21.42 | 20.43 | 18.18 | 17.49 | 19.85 | 16.77 | 17.21 | **15.60** | 15.67 | 15.76 | 15.45 | **15.29** |
| | | MAPE | 24.51 | 21.51 | 21.12 | 20.19 | 18.91 | 17.15 | 19.31 | 16.30 | 18.27 | **16.08** | 16.14 | 16.24 | 15.98 | **15.77** |
| PEMS4 | 60 min | RMSE | 36.66 | 44.56 | 43.92 | 41.27 | 38.12 | 35.55 | 39.70 | 31.88 | 33.34 | 31.35 | **31.21** | 31.31 | 30.68 | **30.59** |
| | | MAE | 23.75 | 28.70 | 28.45 | 26.88 | 24.70 | 22.70 | 25.45 | 19.83 | 20.93 | 19.60 | **19.45** | 19.67 | 19.05 | **19.01** |
| | | MAPE | 18.09 | 19.20 | 18.90 | 17.95 | 17.12 | 14.59 | 17.29 | 13.02 | 14.06 | 12.90 | **12.89** | 12.95 | 12.76 | **12.68** |
| PEMS8 | 60 min | RMSE | 36.33 | 36.16 | 35.34 | 33.62 | 27.83 | 27.83 | 31.05 | 26.22 | 26.70 | 24.53 | **24.39** | 24.67 | 23.99 | **23.63** |
| | | MAE | 23.46 | 23.25 | 22.76 | 21.59 | 17.86 | 18.02 | 19.13 | 16.64 | 16.97 | 15.60 | **15.44** | 15.63 | 14.90 | **14.72** |
| | | MAPE | 15.42 | 14.64 | 14.51 | 14.10 | 11.45 | 11.40 | 12.68 | 10.60 | 11.32 | 10.53 | **10.42** | 10.63 | 10.19 | **10.01** |
| Count | | | 0 | 0 | 0 | 0 | 0 | 0 | 0 | 0 | 0 | 8 | 13 | 6 | 0 | **27** |

more comprehensively model spatio-temporal sequences. At the same time, AutoST$_2$ beats AutoST$_1$ on every task. It shows that using a single-layer but complex internal connection searching method for these tasks is more effective.

Moreover, we can see that the best methods among all baselines are STFGNN, the only method in spatio-temporal synchronous modeling. Comparing GMAN and Graph WaveNet, the two representative methods in spatial-first modeling and temporal-first modeling, respectively, we can see that although they beat other baselines, while our proposed Uni- series model shows a better performance. From this perspective, the proposed method matches the design.

## 5.3  Ablation Study

In AutoST, we design to use gradient-based network architecture search methods to optimize the connections of the modeling modules, and we choose DARTS and PAS as the NAS methods. At the same time, we also conduct experiments to compare the method of random search and AutoSTG [17], which also uses the network architecture search technology for ST sequence prediction. The results are shown in Table 2. We can conclude that our proposed AutoST can outperform AutoSTG with all searching methods. That is because AutoSTG uses more fine-grained temporal convolution and graph convolution structure as the candidates, while AutoST's candidate layers are designed to model the three different spatio-temporal dependencies, which can make better use of the modeling ability of the three layers on different temporal and spatial relations. It reduces the searching space and the search time simultaneously so that AutoST can make better efficiency and accuracy.

## 5.4  Result Visualization

**Forecasting Visualization:** We randomly selected one day from the PEMS-BAY dataset and compared our methods' forecasting results and baseline methods in a visualization way. Typical results are shown in Fig.8, the time span is selected from 13:00 to 19:00, which can represent the most typical scene of the rush hour from afternoon to evening. From the forecasting lines compared to the ground truth, we can see that our proposed methods outperform the two typical baseline methods mostly when forecasting smooth traffic and traffic jam. The bar lies on the bottom of the figures is the forecasting error of each method. We can find that all methods encountered accuracy drop

Table 2: The results of network searching methods.

| Method | METR-LA (60 min) | | | PEMS-BAY (60 min) | | |
|---|---|---|---|---|---|---|
| | RMSE | MAE | MAPE | RMSE | MAE | MAPE |
| AutoSTG | 7.27 | 3.47 | / | 4.38 | 1.92 | / |
| $AutoST_1$-R | 6.12 | 2.97 | 8.32 | 3.69 | 1.62 | 3.65 |
| $AutoST_2$-R | 6.08 | 2.92 | 8.26 | 3.68 | 1.62 | 3.66 |
| $AutoST_1$-D | 6.09 | 2.85 | 7.96 | 3.74 | 1.68 | 3.72 |
| $AutoST_2$-D | 6.16 | 3.25 | 8.48 | 3.64 | 1.60 | 3.61 |
| $AutoST_1$-P | 6.03 | 2.85 | 7.88 | 3.60 | 1.57 | 3.52 |
| $AutoST_2$-P | **6.00** | **2.78** | **7.79** | **3.58** | **1.52** | **3.48** |

(a) The search results of $AutoST_1$. (b) The search results of $AutoST_2$.

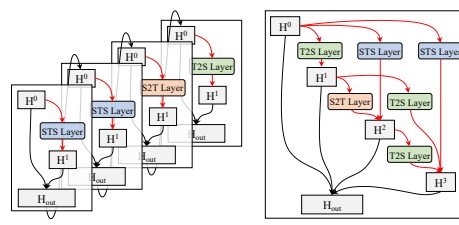

Figure 7: The learned architectures of AutoST.

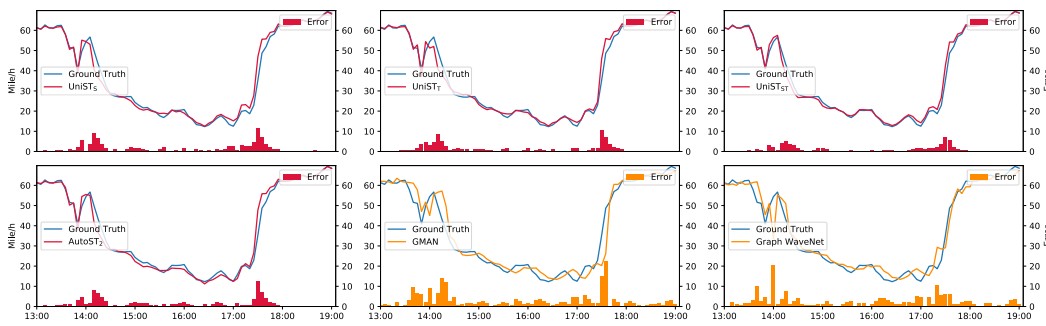

Figure 8: The forecasting visualization in the rush hour of PEMS-BAY dataset. The red and yellow bars on the bottom show the difference between the result and the ground truth at each time step.

around 14:00 and 17:30. That is because the status of the road is changed dramatically. However, our methods are more stable with the changes and can quickly be adapted to the new road state. For the complete data visualization of the day, please refer to Fig. 9 in the appendix.

**Learned Architecture Visualization:** The learned architectures of the $AutoST_1$ and $AutoST_2$ on PEMS-08 dataset are shown in Fig. 7. We can find that although $UniST_T$ has a better effect when modeling with single type modeling module, the search result of $AutoST_1$ show that the *STS Layer* occupies a larger number, and the model achieves better performance than $UniST_T$. This demonstrates that combining and stacking multiple spatio-temporal dependency modeling methods reasonably can better fit the real spatio-temporal dependencies.

In addition, we can find that $AutoST_2$ obtains a complex connections between the four computation nodes in a single cell. And this learned architecture helps $AutoST_2$ achieve state-of-the-art on this task. Although we cannot yet explain why stacking the modules leads to better results, we can see a potentially broad range of applications [25] for unified architecture searching in this way.

## 6  Conclusion

In this work, we illustrated the existence of the modeling gap problem, especially the modeling order, in the spatio-temporal analysis. Moreover, we build three different layers, namely S2T, T2S, and STS, as new network modeling backbones. Then, an automatic searching strategy is proposed to search the optimal modeling priority automatically. Extensive experiments on five real-world datasets show the overwhelming performance over SOTA baselines.

## Acknowledgments

This work was supported by grants from the Natural Science Foundation of China (U20B2053, 62202029) and Foshan HKUST Projects (FSUST21-FYTRI01A, FSUST21-FYTRI02A). Thanks for computing infrastructure provided by Beijing Advanced Innovation Center for Big Data and Brain Computing. This work was also sponsored by CAAI-Huawei MindSpore Open Fund. We gratefully acknowledge the support of MindSpore, CANN (Compute Architecture for Neural Networks) and Ascend AI Processor used for this research.

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
