# A  The Embeddings

In this section, we briefly introduce the four kinds of emebddings consists the fusion embedding.

**Positional embedding**. The goal of position embedding module is to calibrate the position of each time point in the sequence so that the self-attention mechanism can recognize the relative positions between different time points in the input sequence. The positional embedding method we use is the same as [9]: $\mathbf{E}_P(pos, 2i) = \sin(pos/10000^{2i/d})\mathbf{E}_P(pos, 2i+1) = \cos(pos/10000^{2i/d})$, where $pos$ refers to the position in the input sequence, $i \in [0, ..., d/2]$ denotes to embedding feature dimension, and $d$ denotes the feature dimensions of the layer modules.

**Token embedding**. We design the token embedding module in order to enrich the features of each time point by fusion of other features from the adjacent time points within a certain interval. We use $Conv1D$ operation along the time dimension to perform token embedding [31], thus providing more effective information and reference for the subsequent computation of the self-attention mechanism:

$$\mathbf{E}_V = \text{Conv1D}(\mathbf{X}) \quad . \tag{3}$$

**Spatial embedding**. The role of spatial embedding is to locate and encode the spatial locations of different nodes, by which each node at different location possesses a unique spatial embedding. Thus, it enabling the model to identify nodes in different spatial and temporal planes after the dimensionality is compressed in the later computation. The spatial embedding module stores a learnable vector for each differently numbered node as follows:

$$\mathbf{E}_S = \text{Embed}(nodes) \quad , \tag{4}$$

where $nodes$ refers to the list of node numbers. [25] also introduces similar spatial embeddings. For $N$ nodes, $\mathbf{E}_S \in \mathbb{R}^{N \times d}$.

**Temporal embedding**. Temporal embedding aims to map the temporal attributes inherent to the ST sequence into a matrix that serves as an informative complement to the original data. The temporal features that can be extracted from the timestamp include the day-of-week, the time-of-day, and the hour, minute, and other features of the time point. The various temporal features are normalized by $value' = value/\max(value) - 0.5$ scaling it to the range [-0.5, 0.5]. The normalized temporal features are then concatenated as a matrix and projected by another learnable matrix to obtain the corresponding temporal embedding:

$$\mathbf{E}_T = (\text{concat}[T_{DoW}, T_{ToD}, T_{Hour}, T_{minute}]\mathbf{W}_T) \quad , \tag{5}$$

where $T_{DoW}, T_{ToD}, T_{Hour}, T_{minute}$ represents different normalized temporal features that indicate the day-of-week, the time-of-day, the hours, and the minutes, respectively. $\mathbf{W}^T \in \mathbb{R}^{d \times 4}$ denotes a learnable projection matrix.

# B  Experiment Details

| Dataset | Task | Spatial info. | # Sensors | # Time slices | Time span |
|---------|------|---------------|-----------|---------------|-----------|
| METR-LA | speed | distance | 207 | 6519002 | 2012/03/01 - 2021/06/30 |
| PEME-BAY | speed | distance | 325 | 16937179 | 2017/01/01 - 2017/03/31 |
| PEMS-03 | flow | connectivity | 358 | 9382464 | 2018/09/01 - 2018/11/30 |
| PEMS-04 | flow | connectivity | 307 | 5216544 | 2018/01/01 - 2018/02/28 |
| PEMS-08 | flow | connectivity | 170 | 3035520 | 2016/07/01 - 2016/08/31 |

Table 3: Statistics of the datasets.

## B.1  Data Preprocessing

Given the original dataset, we first conduct data preprocessing to build the spatial relationship, normalize the inputs and split the dataset for training and testing.

**Spatial relationship building**. We organize the given relationships in the datasets as graphs. (1) For datasets which providing distance information between nodes, we construct the adjacency matrix

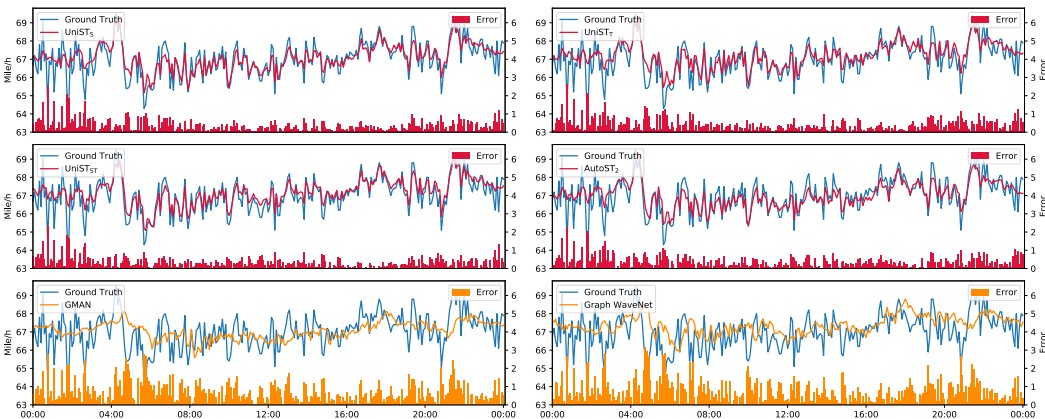

Figure 9: The forecasting visualization in one day of PEMS-BAY dataset.

using a thresholded Gaussian kernel [13] as shown in the formula: $\mathbf{A}_{ij} = \exp\left(-\frac{\text{dist}(v_i,v_j)^2}{\sigma^2}\right)$ if $\text{dist}(v_i, v_j) < k$, else $\mathbf{A}_{ij} = 0$, where $\text{dist}(v_i, v_j)$ represents the distance between node $i$ and node $j$, $k$ denotes the threshold. (2) For dataset with only connectivity information between nodes, we use the connectivity matrix to construct an adjacency matrix. If there is a direct connection between node $i$ and node $j$, $\mathbf{A}_{ij}$ will be set to 1, and if the two nodes are not directly connected, it will be set to 0.

**Data normalization**. We normalize the ST sequence using the *Z-score* normalization so that the model can be better trained and converged, and its definition is shown as: $X' = \frac{X - \bar{X}}{\sigma}$, where $\bar{X}, \sigma$ denotes the mean and standard deviation of the original data.

**Dataset split**. We first split the sequence data in chronological order, taking the first 70% of the original complete sequence as the training set, the next 10% as the validation set, and the last 20% as the test set. After the splitting of each part, the long sequence needs to be divided into multiple subsequences with fixed lengths, which are served as model inputs and labels. The final train/validation/test set consists of these fixed length subsequences. Among them, random operations can be performed to disrupt the order of the subsequences for the training set and the validation set, and the order of the subsequences is maintained for the test set.

## B.2 Spatio-temporal Sequence Forecasting Task Design

In order to fully test the forecasting ability of the model for short-term, medium-term and long-term sequences variation and the effectiveness of spatio-temporal representation learning, we conduct experiments on two traffic speed datasets, METR-LA and PEMS-BAY to predict the data variation of each sensor in the next 15 minutes, 30 minutes and 60 minutes. In order to test the model's ability to learn and forecast the long-term dependence of the ST sequence, we conduct experiments to predict the data variation of each sensor in the next hour on three traffic flow datasets PEMS-03/04/08.

### B.2.1 Metrics

The evaluation metrics we use are Root Mean Square Error (RMSE), Mean Absolute Error (MAE), and Mean Absolute Percentage Error (MAPE). They are calculated as follows:

$$\text{RMSE}(\hat{Y}, Y) = \sqrt{\frac{1}{M}\sum_{i=1}^{M}\left(\hat{Y}_l - Y_i\right)^2},$$

$$\text{MAE}(\hat{Y}, Y) = \frac{1}{M}\sum_{i=1}^{M}\left|\hat{Y}_l - Y_i\right|, \tag{6}$$

$$\text{MAPE}(\hat{Y}, Y) = \frac{100\%}{M}\sum_{i=1}^{M}\left|\frac{\hat{Y}_l - Y_i}{Y_i}\right|.$$

where $M$ denotes the total sample number.

### B.2.2 Baseline Methods

We compared a variety of models and algorithms including statistical methods, traditional machine learning methods, commonly used deep learning methods, and the latest researches.

**VAR** [18]: Vector Autoregressive Model, a statistical model for multi-sequence forecasting.

**SVR** [4]: Support Vector Regression, a machine learning model for regression forecasting using linear support vector machines.

**ARIMA** [2]: Autoregressive Differential Moving Average Model, a statistical model for time series forecasting.

**WaveNet** [20]: A fully connected convolutional network for processing audio sequence data.

**DCRNN** [13]: Diffusion convolution is introduced into the recurrent neural network kernel, which enables the network to deal with spatial relationships and realize the ST sequence forecasting.

**STGCN** [26]: Combines 1D convolution with graph convolution and uses a sequence-to-sequence structure to process ST sequence data.

**Graph WaveNet** [23]: Using dilated 1D convolution to obtain the time-series dependencies within the sequence, and use the adaptive matrix to enhance the spatial relationship. And it uses diffusion convolution to obtain the ST sequence forecasting. It is represented by G.WaveNet in the table of experimental results.

**STFGNN** [12]: Simultaneous acquisition of local and global spatio-temporal correlations is achieved by fusing a dilated convolutional neural network with gating mechanism and a spatio-temporal fusion graph module.

**GMAN** [29]: First obtain the graph representation by performing random walks on the graph, and then obtain the spatio-temporal features separately through the Encoder-Decoder structure similar to the Transformer model using independent temporal and spatial attention modules, and then the ST sequence representation is obtained by gating network fusion.

### B.3 Experiments on Computation Resource Consumption

### B.3.1 Running Time

We compare the training and inference time consumption of our proposed method with various deep learning methods. The experiment is conducted on PEMS-BAY dataset. The speed of baseline methods are from their respective published papers, or calculated with their recommended parameters. The proposed methods all use a 3-layer model structure, and the dimension of each core vector representation is set to 32. The experiments recorded the time for the model to complete one epoch in the training phase, including forwarding and backpropagation, and the time to complete the entire prediction on the test set. The results are shown as lines in Fig.10. Experiment results show that compared with the GMAN model that also uses the self-attention mechanism and the state-of-the-art method STFGNN, the model proposed in this paper has greatly improved the training speed, and is second only to the STGCN model in the overall training speed. In terms of inference speed, since both the STGCN model and the DCRNN model use a step-by-step dynamic decoding method, it will cause accumulated errors and increase the time overhead. Our proposed model performs better inference speed, second only to the Graph WaveNet and GMAN models.

For the two types of AutoST, we report the overall searching time in Table 4. Compared to timings in Fig.10, AutoST1 and AutoST2 take much longer time due to the search process, but a few hours are still acceptable. In real-world application, there are commonly two situations: (1) For a known task with new data, the model can directly use the former searched model architecture, and AutoST have comparable timing as other baselines. (2) For a new task, the model does not know the spatio-temporal model architecture, and needs to search it once.

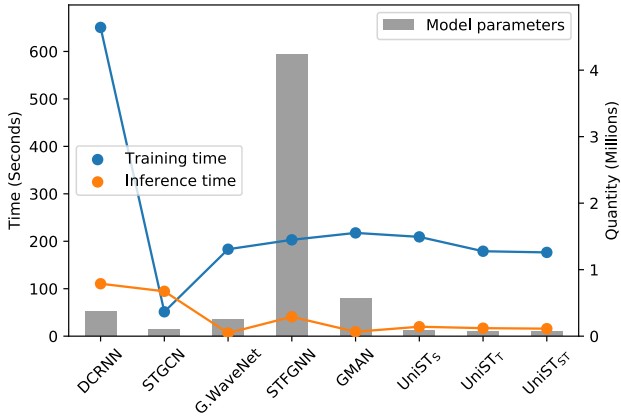

Figure 10: The training/inference time (lines) and the parameter number (bars) of the models.

Table 4: The searching time of AutoST.

| Search Time (h) | AutoST$_1$ | AutoST$_2$ |
|---|---|---|
| METR-LA (60min) | 1.50 | 2.76 |
| PEMS-BAY (60min) | 3.74 | 5.97 |
| PEMS-03 (60min) | 1.63 | 3.01 |
| PEMS-04 (60min) | 0.87 | 1.53 |
| PEMS-08 (60min) | 0.45 | 0.88 |

### B.3.2 Parameter Number

We also conduct experiments on METR-LA and PEMS-BAY datasets to count the parameter numbers of each model. The results of comparison methods are acquired from published paper/code, or use their recommended setting to conduct statistics. The results are shown as bars in Fig. 10. The results show that our proposed model has a relatively more minor number of parameters, second only to the STGCN model, but far outperforms STGCN and other methods. Compared with the GMAN model that also uses the self-attention mechanism, the methods proposed in this paper reduce the number of parameters by up to 89%. Compared with the state-of-the-art method STFGNN, the proposed methods reduce up to 98% parameter.

## C   Limitations and Potential Negative Impacts

Although our AutoST achieved best forecasting performance, the architecture search process is time-consuming. Therefore, AutoST is more suitable for scenarios with static graph such as traffic and body pose, and is inefficient for dynamic graphs like social networks. In the future, we will investigate efficient AutoST for dynamic graphs based on incremental learning and transfer learning. A potential negative impact could be the privacy management in the traffic data.