# OpenReview forum: "AutoST: Towards the Universal Modeling of Spatio-temporal Sequences"
_NeurIPS.cc/2022/Conference — NeurIPS 2022 Accept_

### Official Review · Reviewer_dPZy · 2022-07-10

**Rating:** 5
**Confidence:** 4
**Soundness:** 4 excellent
**Presentation:** 4 excellent
**Contribution:** 3 good

**Summary:**

Two Network architecture search algorithms are proposed, called Automated Spatio-Temporal
modeling approach, AutoST1 and AutoST2. They are focused on automatically crafting spatio temporal forecasting systems. Both the search algorithms operate by selecting from a layer candidate set,  formed by three types of layer: T2S Layer (Temporal-first Modeling Layer), S2T Layer (Spatial-first Modeling Layer), STS Layer(Spatial-temporal Synchronous Layer). All of these three layers have exactly same inputs and outputs,  and operate by assembling in different fashion a Time Series Linear Self-Attention Unit and a High-order Mix Graph Convolution Unit. AutoST2 is the most performing one, which creates a serial architecture of hidden layers, each one of them can be a T2S, S2T, STS layer. Results on traffic datasets support the search algorithm on five traffic forecasting datasets.


**Questions:**

- See my question on the time complexity.
- Having five datasets of traffic forecasting, most of them having very similar structure and content, is definitely a cons. It would have been definitely better to take into account a dramatically different scenario/task (eg pose forecasting)


**Ethics Review Area:**

["Privacy and Security (e.g., consent)", "Legal Compliance (e.g., GDPR, copyright, terms of use)"]

**Limitations:**

No limitations or potential negative impacts have been taken into account by the authors. I think it should be, since managing traffic data could be very impactful on the privacy management. ..

**Strengths And Weaknesses:**

Strengths: The idea of learning an architecture which understands whether spatial or temporal relations are more important to forecast a given signal is intriguing. Figure 2 has a very effective visualization, which makes the intuition of the authors crystal clear.



Weaknesses:
-- The experiments focus on very specific datasets, all of them representing the same problem (traffic forecasting). It would have been definitely better to have more heterogenous tasks, as for example pose forecasting, where the spatial part is the body skeleton (expressed as a graph) and the temporal part let the body joints move accordingly to anatomical costraints. An interesting comparison would have been
Theodoros Sofianos, Alessio Sampieri, Luca Franco and Fabio Galasso  Space-Time-Separable Graph Convolutional Network for Pose Forecasting In Proc. International Conference on Computer Vision (ICCV) Virtual, October 2021
They use a Space-Time Separable GCN (STS-GCN. Which is *not* the same as the STSGCN cited in the proposed paper as [19]).

-- The Complexity of the AutoST1, and especially AutoST2 is not detailed. If complexity cannot be given, timings become very important.  In particular, since AutoST2 has to look for multiple optimal layers at each cell, what is the time spent overall, and for completing each layer of Fig.5.b?

-- The definition of Adaptive Diffusion Convolution is clear (ChebNet + ADC), but its motivation is not. A sentence on the rationale for this choice would be beneficial.

-- In section 3.2, it is not clear in the result connection formula what is o^{(i,j)}, and why i<j, w.r.t. which ordering?

-- The rationale why T2S, S2T, STS have been built as described in 4.2.1, 4.2.2., 4.2.3 ,with that precise architecture, is not given.

MINOR:

--Fig:3 S2T Layer: firstly model temporal... -->T2S Layer: firstly model temporal...
--pag.3: is automated methods --> are automated methods
--pag.3 our proposed trainning framework --> our proposed training framework
--In the objective function of section 3.2, w and alpha are undefined.

---

> ### Author Response · Authors · 2022-08-02
> **Response to reviewer dPZy**
>
> Thanks for your valuable advice and for appreciating our idea and paper writing.
>
> > The experiments focus on very specific datasets. It would have been definitely better to have more heterogenous tasks, as for example pose forecasting.
>
> Thank you for suggesting this. We agree that adding a different scenario will further strengthen the contribution of this paper, although we already have 5 datasets on traffic forecasting. We have been running the new experiments, but the results are not complete yet, and we will update them later during the rolling review. Due to the space limitations of the paper, we plan to add the new results in the appendix. Furthermore, compared to the body pose forecasting task, the traffic forecasting tasks are more challenging because it has more nodes in the graph (~300 vs ~20) and a longer time span (60min vs 1s). It is promising that our method can also handle this scenario.
>
> > The Complexity of AutoST1, and especially AutoST2 is not detailed. If complexity cannot be given, timings become very important.
>
> Thank you for pointing this out. We have added this to the appendix Table 3. The detailed timings of AutoST1 and AutoST2 are:
>
> | Search Time (h) | AutoST_1 | AutoST_2 |
> | :---: | :---: | :---: |
> | METR-LA | 1.50 | 2.76 |
> | PEMS-BAY |  3.74 | 5.97 |
> | PEMS-03 | 1.63 | 3.01 |
> | PEMS-04 | 0.87 | 1.53 |
> | PEMS-08 | 0.45 | 0.88 |
>
> Compared to timings in Figure 10, AutoST1 and AutoST2 take much longer time due to the search process, but a few hours are still acceptable. In real-world application, there are commonly two situations: (1) For a known task with new data, the model can directly use the former searched model architecture, and AutoST has comparable timing as other baselines. (2) For a new task, the model does not know the spatio-temporal model architecture and needs to search it once.
>
>
> > The definition of Adaptive Diffusion Convolution is clear (ChebNet + ADC), but its motivation is not. A sentence on the rationale for this choice would be beneficial.
>
> We have added the rationale of ChebNet + ADC in Sec 4.1.2: "ChebNet focuses on 1st-order neighbor information, while AdapDC focuses on multi-hop information". We design this module for better spatial information mixing and feature extraction. It builds a high-order mix graph convolution where each order has a dominating neighbor information and multi-hop spatial information. Meanwhile, the experimental results demonstrated this design is effective.
>
> > In section 3.2, it is not clear in the result connection formula what is $o^{(i,j)}$, and why $i<j$, w.r.t. which order?
>
> Thanks for pointing this out. We have explained it in Definition 1, Page 6, and we will emphasize it in section 3.2: "where $o^{(i,j)}$ is an operator, e.g. layers in a model, represented by a directed edge from node $i$ to node $j$". $i, j$ are indexes of the nodes. The equation means that the successor node j is the sum of the operation results of all predecessor nodes i. For instance, as illustrated in Figure 5(b), the value of node $H_4 = o^{(0,4)} (H_0) + o^{(1,4)} (H_1) + o^{(2,4)} (H_2) + o^{(3,4)} (H_3)$.
>
> >The rationale why T2S, S2T, and STS have been built as described in 4.2.1, 4.2.2, 4.2.3, with that precise architecture, is not given.
>
> We will add the rationale for the architecture design of T2S, S2T, and STS in the revised paper. Here are the rationales. "The block in green "HighOrder MixGC" and in orange "Linear Self-Attention" is responsible for capturing spatial information and temporal information, respectively. Inspired by Transformer, we design the encoder layers as follows: For S2T: $K$, $V$ involves spatial information, the attention is calculated between $K$ and the inputs $Q$ to capture the temporal information based on the representation $V$. And we achieve spatial to temporal modeling. Similarly and simpler, the T2S layer performs self-attention first to modeling temporal dependencies, and then modeling spatial, while the STS layer performs spatial and temporal simultaneously."
>
> > No limitations or potential negative.
>
> Thanks for your suggestion. We have added a limitation section in the revised appendix due to the page limit: "Although our AutoST achieved best forecasting performance, the architecture search process is time-consuming. Therefore, AutoST is more suitable for scenarios with static graphs such as traffic and body pose and is inefficient for dynamic graphs like social networks. In the future, we will investigate efficient AutoST for dynamic graphs based on incremental learning and transfer learning. A potential negative impact could be the privacy management in the traffic data."

---

> > ### Comment · Reviewer_dPZy · 2022-08-08
> > **Thanks for the rebuttal**
> >
> > The rebuttal clarified amny aspects, thanks. Also, I appreciated the answer to Reviewer MfGi. I truly hope that experiments on another application eg pose forecasting will show good results for the rolling review. I'm moderately positive about this paper, and if the approach shows to be effective on more applications, I can push strongly for acceptance.

---

> > > ### Author Response · Authors · 2022-08-09
> > > **Thanks to the Reviewer dPZy**
> > >
> > > Thank you for the encouraging reply. We are trying our best to work on the new experiment. We will update the result soon.

---

> > > ### Author Response · Authors · 2022-08-09
> > > **Response to Reviewer dPZy**
> > >
> > > We would like to thank all the reviewers for the constructive feedback and your patience in waiting for our extended experiments. We appreciate Reviewer dPZy's suggestion and encouragement on the experiment.
> > >
> > > We tried our best in the past few days to apply our proposed UniST / AutoST to the human pose prediction application, which is an area we haven't been in before. The results satisfied us, our methods show competitive performance compared with the SOTA methods.
> > >
> > > For the Human3.6M dataset, we chose the longest prediction period (400ms and 1000ms) in both the short-term prediction task and the long-term prediction task, which we think can best reflect the performance of the model among many tasks.
> > >
> > > **Table 1: MPJPE error in mm of 3D joint positions on Human3.6M.**
> > >
> > > |             | Walking |      | | Eating |      | |  Smoking |      | |  Discussion |       |
> > > |-------------|---------|------|----| -------|------| ----| ---------|------| ----| ------------|-------|
> > > | *msec*      | *400*   |*1000*| | *400*  |*1000*|  | *400*   |*1000*|  | *400*      |*1000* |
> > > | DCT-RNN-GCN | 39.8    | 58.1 | | 36.2   | 75.5 |  | 36.4    | 69.5 |  | 65.4       | 119.8 |
> > > | STS-GCN     | 32.9    | 51.8 | | 25.4   | 52.4 |  | 25.8    | 50.0 |  | 40.2       |**78.8**|
> > > | UniST_S     | 33.1    | 50.2 | | **24.2**| 50.9 |  | 25.2    | 47.8 | | **38.2**    | 79.4  |
> > > | UniST_T     | 32.4    | 50.1 | | 25.6   |**49.3**| | **24.5**|**47.3**| | 39.5     | 80.1  |
> > > | AutoST_1    | 31.7    | 49.0 | | ---    | ---  |  | ---     | ---  |  | ---        | ---   |
> > > | AutoST_2    | **31.5**|**48.2**| | ---    | ---  |  | ---     | ---  |  | ---        | ---   |
> > >
> > >
> > > For the AMASS dataset, we chose three representative sub-tasks: 80ms (for short-term), 400ms (for medium-term), 1000ms (for long-term).
> > >
> > > **Table 2: MPJPE error in mm of BMLrub test sequences of AMASS.**
> > >
> > > |             | AMASS |      |      |
> > > |-------------|-------|------|------|
> > > | *msec*      | *80*  | *400*|*1000*|
> > > | DCT-RNN-GCN | 11.3  | 42.0 | 67.5 |
> > > | STS-GCN     | 10.0  | 24.5 | 45.5 |
> > > | UniST_S     | 10.4  | 23.2 | 45.7 |
> > > | UniST_T     | 10.2  |**23.0**|**45.2**|
> > > | AutoST_1    |  9.9  | ---  | ---  |
> > > | AutoST_2    |**9.7**| ---  | ---  |
> > >
> > > The results show evidence that our proposed methods can be finely generalized to other spatial-temporal prediction applications (on "Human3.6M" and "AMASS", the 3rd dataset "3DPW" in [1] is also under experiment).
> > >
> > > [1] Theodoros Sofianos, Alessio Sampieri, Luca Franco, and Fabio Galasso, Space-Time-Separable Graph Convolutional Network for Pose Forecasting, In Proc. International Conference on Computer Vision (ICCV), Virtual, October 2021.
> > >
> > > Unfortunately, with limited time, we cannot run all models and all settings of the experiment, even for sufficient parameter tuning. But we will keep working on this in the next few days. And we will surely complete and include all of this part of experiment in the final version of our paper. The code about human pose experiments will be also published.
> > >
> > > Thank all the reviewers again for the constructive suggestions that helped us to improve our work, we really enjoy reading and learning from your comments.

---

### Official Review · Reviewer_2S7t · 2022-07-13

**Rating:** 8
**Confidence:** 3
**Soundness:** 4 excellent
**Presentation:** 3 good
**Contribution:** 4 excellent

**Summary:**

Paper proposes a novel universal framework for spatio-temporal forecasting. The framework combines spatial first, temporal first and spatio-temporal synchronous approaches in one network. These are implemented as S2T Layer, T2S Layer and STS Layer respectively. This is termed UniST.

Paper further proposes a novel automatic search strategy by network architecture search.

Experimental results on 5 datasets across 9 tasks outperforms existing methods significantly.

**Questions:**

No question.

**Limitations:**

Not applicable.

**Strengths And Weaknesses:**

Strengths
1. Paper is well-written and easy to understand.
2. Method is presented clearly with detailed explanations and good use of diagrams.
3. Experiments are extensive and strongly supports the main claims of the paper

Weakness
1. Minor. The labeling of Table 1 is too brief. It should contains more information to explain the Table content.

---

> ### Author Response · Authors · 2022-08-02
> **Response to Reviewer 2S7t**
>
> We are very glad to know that you enjoyed reading our paper! We thank your positive reviews on our research question, method, and experiment, which are very encouraging.
>
> > The labeling of Table 1 is too brief. It should contain more information to explain the Table content.
>
> We have fixed this in the revised version. The caption of Table 1 is now: "ST sequence forecasting performance. The bold and shaded numbers are the best results of all methods. The bold numbers are the best results of manually designed models".
>
> > Limitations: Not applicable
>
> Due to the page limit, we have added limitations in the revised appendix. "Although our AutoST achieved the best forecasting performance, the architecture search process is time-consuming. Therefore, AutoST is more suitable for scenarios with static graphs such as traffic and body pose and is inefficient for dynamic graphs like social networks. In the future, we will investigate efficient AutoST for dynamic graphs based on incremental learning and transfer learning. A potential negative impact could be the privacy management in the traffic data."

---

### Official Review · Reviewer_MfGi · 2022-07-15

**Rating:** 6
**Confidence:** 3
**Soundness:** 2 fair
**Presentation:** 3 good
**Contribution:** 2 fair

**Summary:**

The paper focuses on architecture design for spatio-temporal modeling for forecasting tasks. The paper first proposes three types of layers (i.e., S2T, T2S, STS) which have emphasis on modeling spatial, temporal and spatio-temporal information, respectively. The three layers are then applied in a unified encoder-decoder framework for spatio-temporal forecasting tasks. An automated architecture search algorithm is further adopted to better decide the connection configuration of different modeling layers. The final model, AutoST, achieves better results than baseline methods as well as the proposed UniST, which only involves one of the three modeling layers (S2T / T2S / STS).

**Questions:**

Minor:
1. A comma is missing in the last equation of Eqn. 1.
2. The equation in L204 is inconsistent with the text "the outputs of each extractor are added to form the final output".


**Limitations:**

Not discussed in the paper.

**Strengths And Weaknesses:**

Strength
1. The idea of analyzing and unifying different types of spatio-temporal modeling (i.e., spatial first, temporal first, spatio-temporal synchronous) is well-motivated and potentially valuable to the research community.
2. The paper provides clear ablation studies (in Table1) that show (1) the performance difference of the three modeling types on different datasets / temporal setups; (2) the effectiveness of introducing automated architecture search to fuse the three types of modeling layers.

Weaknesses
1. Although I like the idea of unifying different types of spatio-temporal modeling, the technical novelty and contribution of this work is not significant enough. Most of the techniques in the paper are adopted from prior works with minor modifications, for example, linear self-attention, mix graph convolution (with combination of ChebNet and AdapDC), and DARTS / PAS for automated architecture search. Even from the framework design perspective, the ideas of spatial-first / temporal-first and joint spatio-temporal modeling are widely explored in previous work mentioned as baselines in Table 1, as well as the decoupled spatio-temporal modeling design [1,2] for video action recognition. The encoder-decoder framework adopted in UniST is also widely used for both NLP and CV tasks in recent years.

In addition, although the paper provides comparison between UniST and other baseline methods in Table 1, the comparison is not totally fair due to different model complexity, computation cost and even optimization details. This also makes the contribution of the proposed UniST insufficient.

2. The major contribution of this work to me is using architecture search algorithms to unify three types of modeling layers, i.e. the final AutoST model. While the paper compares the results of using different NAS algorithms, an important baseline is missing. Instead of only using one the the three types of modeling layers (UniST), it is intuitive to have a model that includes all of the three layers at each STE and fuses all three outputs, optionally augmented with a gating module to introduce a weighted fusion mechanism. With this baseline result provided, the necessarity of using NAS to select modeling layers would be more convincing.



[1] Tran, Du, et al. "A closer look at spatiotemporal convolutions for action recognition." Proceedings of the IEEE conference on Computer Vision and Pattern Recognition. 2018.
[2] Qiu, Zhaofan, Ting Yao, and Tao Mei. "Learning spatio-temporal representation with pseudo-3d residual networks." proceedings of the IEEE International Conference on Computer Vision. 2017.

---

> ### Author Response · Authors · 2022-08-02
> **Response to Reviewer MfGi**
>
> Thank you very much for your encouraging comments and insightful feedback.
>
>
> > Technical novelty and the contribution is not significant enough.
>
> We clarify the contributions of this paper in the Introduction.  See revised paper.
> We clarify that the main contributions lie in our novel research question which is identifying ST modeling order automatically without prior knowledge of the task or manually designed model framework. As we mentioned in Sec 2, we identified and categorized existing works into 3 types: spatial-first, temporal first, and joint ST modeling. They all require intuition or a deep understanding of the task, manual design of the model framework, and iterations on the framework with different modeling priorities. We build upon these baselines and contribute by first identifying the problem of automatically designing the ST forecasting model and first verifying that AutoST helps with performance.
> We argue that our technical contribution is more than adopting and modifying existing techniques. First, we propose UniST together with 3 modeling units, i.e. S2T, T2S, STS Layers. Compared to existing methods, they can serve as unified and replaceable building blocks with the same dimension of input and output. These 3 layers are designed to be simple but effective so that they can be easily organized into a large framework with AutoST. Experiments also verified that these 3 units (UniST) can outperform baselines. Furthermore, we contribute by formulating the modeling priority issue as a model structure search problem. As for the encoder-decoder framework, it is basic and is not our contribution.
>
> >Unfair comparison due to different model complexity, computation cost, and optimization details.
>
> Figure 10 in the appendix illustrates the parameter number ("model complexity" raised by Reviewer MfGi) and the training/inference cost ("computation cost" raised by Reviewer MfGi)  of our UniST and baselines. For convenience, we list the values in the following table, along with the mean performance of all methods (the mean of the results over all tasks in Table 1, smaller is better), and we calculate a rough Cost-Performance Ratio (CP Ratio = normalize(1 / Mean Performance / (Training+Inference Time * #Parameters) ).
>
> | Metrics | DCRNN | STGCN | G.WaveNet | STFGNN | GMAN | UniST_S | UniST_T | UniST_ST |
> | :----: |  :----:  | :----:  | :----:  | :----:  | :----:  | :----:  | :----:  | :----:  |
> | Train (s) | 650.64 | 51.35 | 183.21 | 203.05 | 217.62 | 209.27 | 179 | 176.51 |
> | Inference (s) | 110.52 | 94.56 | 6.55 | 40.86 | 9.34 | 19.89 | 16.85 | 15.81 |
> | # Parameters (k) | 372 | 99 | 248 | 4241 | 565 | 89 | 68 | 71 |
> | Mean Performance | 10.33 | 11.56 | 9.90 | 9.26 | 10.08 | 8.86 | 8.82 | 9.00 |
> | Cost-Performance Ratio| 0.04 | 0.70 | 0.25 | 0.01 | 0.09 | 0.65 | 1.00 | 0.96 |
>
> We think the comparison is totally fair. For the "optimization details" raised by Reviewer MfGi, we ran the source code provided by the original paper and strictly follow the recommended hyperparameter settings and the optimization settings in the original paper, or directly using the experiment results which reported in the original paper. We guarantee each model was finely converged. Under this background, our UniST achieves state-of-the-rt even with relatively less model complexity. The Cost-Performance Ratio shows how efficient our UniST is when reaching SOTA. Its model parameters are several times or even dozens of times lower than those of the previous SOTA method while achieving a 5%~24% performance improvement.
>
> >Missing baseline: fusing three types of layers.
>
> We had tried module fusion to combine the output of 3 different layers before. The result of the model was between the best UniST and the worst UniST. One possible explanation is that when the spatio-temporal modeling priority of a task is exactly one of the three types, say spatial first, then combining the output of the other two layers (temporal first and spatio-temporal synchronous) may harm the final model output. This motivated us to propose AutoST.
> A baseline similar to your idea is AutoST-R in an ablation study, where the three layers are randomly chosen and combined according to the two AutoST templates. The results in Table 2 are the best of 10 random runs. This means that not every combination of the 3 layers works and justifies the necessity of AutoST.
>
> >The equation in L204 is inconsistent with the text.
>
> Thanks for pointing this out. We have modified it to: $ \mathbf{X_{en}} = \sum_{i=1}^{L}{\operatorname{STE}^{i}(\mathbf{X}_0)} $, where $ L $ refers to the number of ST extractors.
>
> > Limitation not discussed.
>
> We have added the limitations in the revised appendix.

---

> ### Author Response · Authors · 2022-08-09
> **[Message to Reviewer MfGi] A Gentle Reminder for Author-Reviewer Discussion**
>
> Dear Reviewer MfGi,
>
> Many thanks for reviewing our paper. We really appreciate the opportunity to learn from your constructive suggestions.
>
> Since the author-reviewer discussion will end in one day, could you please let us know if you have any other concerns and questions? We are more than happy to discuss and address them.
>
> Best regards,
>
> Authors

---

### Author Response · Authors · 2022-08-02
**General response**

We thank all the reviewers for appreciating our paper, including our idea (Reviewer MfGi, 2S7t, and dPZy), experiment (MfGi, 2S7t), and paper writing (2S7t and dPZy). And we are very grateful for your insightful comments and valuable suggestions.

- Additional clarification (MfGi, dPZy): We have clarified our contribution, especially the novelty of our research question and technical solution (MfGi). And we also clarified the model complexity and computation cost (MfGi and dPZy).

- Limitation (MfGi, 2S7t, and dPZy): We have added a paragraph on limitations and potential negative impact in the appendix.

- Justification and corrections of minor issues.

*We have submitted the revised paper and appendix.*

---

### Meta-Review · Area_Chair_UbdQ · 2022-08-23

**Recommendation:** Accept
**Confidence:** Certain

**Metareview:**

This paper presents a framework for spatio-temporal sequence analysis.  The framework contains spatial, temporal, and spatio-temporal building blocks and is operationalized in an architecture search approach.  The reviewers considered the authors' rebuttal and engaged in further discussion regarding the merits of the paper.

There are concerns over the overall novelty of the approach and the combination of techniques that are assembled in the method.  However, the potential of the proposed building blocks and overall empirical results were deemed to be a solid contribution.  For these reasons, this paper is recommended to be accepted to NeurIPS.

**Award:**

No

---

### Decision · Program_Chairs · 2022-09-14

Accept